# Eosinophils and Lung Cancer: From Bench to Bedside

**DOI:** 10.3390/ijms23095066

**Published:** 2022-05-03

**Authors:** Anne Sibille, Jean-Louis Corhay, Renaud Louis, Vincent Ninane, Guy Jerusalem, Bernard Duysinx

**Affiliations:** 1Department of Pulmonology, University Hospital of Liège, Domaine de l’Université B35, 4000 Liège, Belgium; jlcorhay@chuliege.be (J.-L.C.); r.louis@chuliege.be (R.L.); bduysinx@chuliege.be (B.D.); 2Department of Pulmonary Medicine, CHU Saint-Pierre, Université Libre de Bruxelles (ULB), 1050 Brussels, Belgium; vincent.ninane@stpierre-bru.be; 3Department of Medical Oncology, University Hospital of Liège, Domaine de l’Université B35, 4000 Liège, Belgium; g.jerusalem@chuliege.be

**Keywords:** eosinophils, non-small cell lung cancer, immunotherapy, biomarkers, predictive value, prognostic value

## Abstract

Eosinophils are rare, multifunctional granulocytes. Their growth, survival, and tissue migration mainly depend on interleukin (IL)-5 in physiological conditions and on IL-5 and IL-33 in inflammatory conditions. Preclinical evidence supports an immunological role for eosinophils as innate immune cells and as agents of the adaptive immune response. In addition to these data, several reports show a link between the outcomes of patients treated with immune checkpoint inhibitors (ICI) for advanced cancers and blood eosinophilia. In this review, we present, in the context of non-small cell lung cancer (NSCLC), the biological properties of eosinophils and their roles in homeostatic and pathological conditions, with a focus on their pro- and anti-tumorigenic effects. We examine the possible explanations for blood eosinophilia during NSCLC treatment with ICI. In particular, we discuss the value of eosinophils as a potential prognostic and predictive biomarker, highlighting the need for stronger clinical data. Finally, we conclude with perspectives on clinical and translational research topics on this subject.

## 1. Introduction

Paul Ehrlich first described eosinophils more than a century ago and already suggested that their alpha-granules contain secretory products [1]. Eosinophils are multifunctional white blood cells (WBC) whose functions have been intensely studied in both physiological and pathological conditions. Their role in non-oncological pulmonary diseases such as asthma and chronic obstructive pulmonary disease (COPD) has been emphasised by major therapeutic developments in the field, more specifically inhaled corticosteroids (ICS) and agents targeting the interleukin (IL)-5 pathway that is essential for the expansion, recruitment, and migration of eosinophils in both physiological and pathological (inflammatory) conditions [2,3]. In oncological diseases also, the study of WBC (neutrophils, lymphocytes and eosinophils) has gained interest, particularly since the advent of immune checkpoint inhibitors (ICI) [4]. In this setting, WBC counts have been studied for their potential prognostic and predictive value in various solid tumors such as non-small cell lung cancer (NSCLC) [5]. Paralleling this, a paradigm shift was observed in the study of solid tumors, highlighting the importance of the tumor microenvironment (TME), which consists of immune and non-immune cells, and of chemo- and cytokines interacting with each other (cross-talk) [6]. Here, we review in the context of NSCLC the biological properties of eosinophils in humans and their roles in homeostatic and pathological conditions, with a focus on their pro- and anti-tumorigenic effects. We also explore possible explanations for blood eosinophilia during NSCLC treatment with ICI. In particular, we discuss the value of eosinophils as a potential prognostic and predictive biomarker, highlighting the need for stronger clinical data. Then, we conclude with suggestions for clinical and translational research topics on this subject.

## 2. Biology of Eosinophils

Eosinophils are granulocytes that differentiate from multipotent stem cells, called common myeloid progenitors in humans and granulocyte/macrophage progenitors in mice [7,8]. According to recent research, the lineage of myeloid cells is set early in the development of different cell subtypes [9]. Mack EA and colleagues reviewed the major transcription factors identified in the eosinophil lineage commitment [10]. They describe the central role of c/EBPα, GATA-1&2, FOG, PU.1, TRIB-1, and IRF8 (Figure 1). Not only the presence of those transcription factors seems important, but also the level and the timing of their expression for eosinophil development. Eosinophil precursors are further matured, expanded, and activated by cytokines, among which IL-5 (in physiological and pathological conditions) and IL-33 (in pathological conditions) play a central role [10]. The major importance of IL-5 has been demonstrated by several experiments where its deletion or overexpression in mice led to eosinophil depletion or excessive synthesis, respectively, and by clinical trials in severe asthma patients displaying a profound eosinophil depletion when treated with IL-5 antagonists, leading to a dramatic control of their symptoms and of the need for oral corticoids [11,12,13]. Interestingly, it is now believed that IL-5 orchestrates the action of other cytokines, such as IL-4, rather than acting as a sole direct trigger on eosinophil precursors via binding to its receptor, IL-5 Receptor unit α (IL-5Rα) [14]. Once triggered, eosinophils are released in a mature state in the blood where they stay for a short time (half-life of 18 h) [15]. In physiological steady-state conditions (see below), eosinophils migrate to the gastrointestinal tract [16] and, to a lesser extent, to the thymus, mammalian gland, and uterus [17,18]. This occurs under the action of chemokine eotaxin-1 (also called CCL11). In inflammatory conditions, the recruitment of eosinophils to alternative tissues such as the lungs is triggered by cytokines (IL-4, IL-5, IL-13, IL-33) [19,20,21,22], adhesion molecules (β-integrins) [23], and eotaxins-1,-2 and -3 (CCL11, CCL24, and CCL26, respectively) [24]. Thus, the expansion and survival of eosinophils depend on IL-5. Eosinophil lung infiltration depends on both IL-5 and on eotaxins. The life span of eosinophils in tissues is shorter in homeostatic conditions [2–5 days] than in inflammatory conditions (~two weeks), at least in vitro [25,26].

Morphologically, eosinophils can be characterized by their intracellular content and by their surface receptors (Figure 2). A bilobed acidophilic nucleus and intracellular granules are common to all species [27]. The granules can be divided into primary granules (containing Charcot–Leyden crystal proteins and lipids), secondary granules, and small granules. In human eosinophils, secondary granules contain four predominant cytotoxic proteins called cationic proteins: major basic protein (MBP)-1, eosinophil peroxidase (EPX), eosinophil cationic protein (ECP), and eosinophil-derived neurotoxin (EDN), the latest two also showing a ribonuclease activity. The granules also contain cytokines, chemokines, and growth factors that enable eosinophils to play their role in inflammation. Cell-surface receptors of eosinophils are numerous [28]. They can be classified into: adhesion molecules (selectins), chemotactic factor receptors (e.g. chemokine receptor 3 (CCR3)), cytokine receptors (e.g., IL-5Rα/β), complement receptors, immunoglobulin receptors, inhibitory receptors (e.g., sialic acid-binding immunoglobulin-like lectin-8 (Siglec-8)), and pattern-recognition receptors (PRR; including Toll-like receptors and RAGE). The PRR recognises danger signals, also called alarmins. These can be of exogenous (infectious) origin (bacterial, fungal, or parasitic; so-called pathogen-associated molecular patterns-PAMPs) or endogenous, tumor-derived signals (so-called danger-associated molecular patterns-DAMPs). Activation of the PRR by the alarmins leads to expansion, adhesion to blood vessels, chemotaxis, degranulation, and cell-to-cell interactions of eosinophils [28], triggering the immune system [29]. IL-33 is an epithelial- and tumor-derived cytokine belonging to the IL-1 cytokine family [30]. It seems to be a crucial alarmin in host defense against tumors. Indeed, eosinophils recruited and activated through IL-33 were shown to be responsible for tumor growth control and for the prevention of pulmonary metastases development in melanoma-bearing mice. Mechanisms leading to these anti-tumorigenic effects have been deciphered and are detailed further. Andreone and colleagues underline the central role of IL-33 through in vitro experiments where induction of eosinophil degranulation by IL-33 in the context of cancer is even superior to that of IL-5 [31].

## 3. Role of Eosinophils in Physiological Steady-State Conditions

Eosinophils are similarly found in various tissues of healthy humans and mice: bone marrow, blood, gastrointestinal tract, thymus, secondary lymphoid tissues, uterus, and adipose tissue. They are implicated in diverse processes, highlighted by the study of IL-5 overexpressing, eosinophil-deficient or cytokine reporter mice [32,33].

The first role of eosinophils is to contribute to tissue development, as is the case in the mammary glands [18], in the uterus [17,34,35], and in the gastrointestinal tract, where they contribute to the development of the Peyer’s patches [16,36]. The second role of eosinophils is in tissue regeneration. As an example, the eosinophil-dependent IL-4 production proved to be crucial for the differentiation of fibrocyte-adipocyte progenitors into hepatocytes and myocytes in the context of liver or muscle injury [37,38]. Thirdly, eosinophils take part in metabolism. In adipose tissue, their IL-4 and IL-13 production leads to the differentiation of macrophages into the M2-phenotype that has greater insulin sensitivity [39] and to the increase in thermogenic, “beige” adipocytes [40]. Finally, eosinophils appear to be of great importance in immune homeostasis, playing a role as innate immune cells and as regulatory cells for the adoptive immunity. Indeed, the priming of B lymphocytes, as well as maintenance of plasma cells within the bone marrow or intestinal mucosa, are (partly) promoted by eosinophil-linked mechanisms: production of IL-4, IL-6, and the activation and proliferation-induced ligand (APRIL) cytokines [41,42,43,44]. Moreover, IgA production, microbiome composition, the integrity of the mucosal barrier, and the development of Peyer’s patches are, in mice at least, all eosinophil-driven through IL-6, APRIL, and transforming growth factor (TGF)-β [36,45]. Lastly, eosinophils are mediators of T-cell tolerance: in the thymus, they participate in the destruction of self-reactive T cells via the secretion of indoleamine 2,3-deoxygenase (IDO) [46].

## 4. Eosinophils and Cancer: The Bench Side

The recruitment of eosinophils at tumor sites relies on tumor cells and on the inflammatory reaction (necrosis) they induce, as well as on peri- or intra-tumoral immune cells (lymphocytes, mast cells, dendritic cells) that can secrete eosinophil chemoattractants [47]. Based on in vitro models of NSCLC, Huang and colleagues demonstrated that eosinophils are attracted by type 2 cytokines (IL-5, IL-4, IL-10, and IL-13) that are produced by tumor cells [48]. GM-CSF and CCL11 (eotaxin 1), which are present in tumor tissue, contribute to the attraction of eosinophils [49,50]. Hollande and colleagues emphasised the role of CCL11 by demonstrating that dipeptidyl peptidase DPP4 (CD26) inhibitor sitagliptin led to enhanced tumor control through enhanced CCL11-mediated eosinophil recruitment at the tumor site [51]. Furthermore, the role of dying tumor cells in eosinophil recruitment was demonstrated in a mouse model for melanoma, where eosinophil concentrations were significantly higher in the capsule (fibrotic area) and in the central (necrotic) area of the lesions [52]. The following alarmins promoting eosinophil infiltration of tumors were identified: high-mobility group box-1 protein (HMGB-1) and IL-33 [30,53]. Recent data on colorectal cancer suggest that the gut microbiota may also influence eosinophil recruitment in such cancers [54].

Preclinical data reveal both anti- and pro-tumorigenic activities of eosinophils, both through direct and indirect mechanisms. As a first step in exploring the hypothetical anti-tumorigenic role of eosinophils, several authors manipulated eosinophil-linked cytokines (IL-4 or IL-33 injections, CCL11, and IL-5 depletion) [30,50,55]. They observed that tumor incidence and/or growth were inversely correlated with eosinophil infiltration. Further in vitro studies showed more precisely the mechanisms by which activated eosinophils can control tumors. In addition to a direct cytotoxic effect on cancer cells through degranulation [30,56], activated eosinophils recruit, activate, and lead to the maturation of several immune cells promoting tumor rejection [30,57,58,59] (Figure 3). Carretero and colleagues showed that activated eosinophils recruit cytotoxic CD8^+^ T cells and are essential for tumor control in their melanoma mouse model [57]. They also demonstrated that eosinophils are capable of macrophage polarisation into an antitumor (M1) phenotype. A pivotal study in colorectal cancer identified that intratumoral eosinophils exert these anti-tumorigenic effects through interferon-gamma (IFNγ) signaling [54]. Additionally, eosinophils tend to normalise tumor vasculature, a crucial factor for tumor maintenance and expansion. Indeed, depletion of eosinophils led to increased vascular leakiness, diminished perfusion, and diminished coverage by mature pericytes [57]. 

However, pro-tumorigenic effects of eosinophils have also been reported. As an example, preclinical models of oral squamous cell carcinoma showed reduced growth when eosinophil infiltration was hampered [60,61]. A model of cervix carcinoma also revealed that eosinophils, activated by tumor-generated thymic stromal lymphopoietin (TSLP), triggered tumor growth [62]. Eosinophils facilitate the recruitment of regulatory T cells (Treg) [63], inhibit cytotoxic T cells via the production of IDO [64], and induce the polarisation of macrophages into the M2, immunosuppressive phenotype through the production of IL-13 [65]. Finally, eosinophils produce many growth factors, with direct effects on tumor growth, metastatic spread, matrix remodeling, or on tumor-associated blood vessels [66].

Those seemingly opposing roles of eosinophils in tumors probably reflect their functional plasticity rather than underline contradictory findings. Firstly, eosinophils are, similar to other myeloid cells, part of the tumor microenvironment (TME), an entity where tumor cells, inflammation, and immune cells interact and evolve over time [67,68]. It is reasonable to think that, as for macrophages and neutrophils, eosinophils’ behavior could vary depending on the surrounding stimuli (cytokines, exosomes) [69,70]. Indeed, while IFNγ and IL-33 trigger an anti-tumorigenic role of eosinophils, IL-5 favors their pro-tumorigenic function [30,54,63]. Secondly, in light of the data described, a differential role for eosinophils according to the histologic subtype might be suspected: immuno-supportive in melanoma, immuno-suppressive in oral squamous or cervix carcinoma. However, it may be so that different tumor types simply reflect different TME. Thirdly, phenotypic studies of eosinophils in asthma mouse models showed eosinophils with different localisations (airway lumen vs. epithelium), morphology (ring-shaped vs. segmented nucleus), and different gene and cytokine expression profiles, reflecting different functions [71,72,73]. This, however, remains to be demonstrated in the context of cancer.

## 5. Eosinophils and Lung Cancer: The Bedside

### 5.1. Blood Eosinophils (B-Eos)

The first data on cancer patients showing an association between anti-neoplastic treatment and eosinophilia came from a cohort of 20 patients treated with IL-2 and lymphokine-activated killer cells for advanced cancer [74]. A study by van Haelst Pisani and colleagues further demonstrated that IL-2 administration was followed by IL-5 production and eosinophilia [75]. Some 20 years later, several authors demonstrated an association between B-Eos, anti-cytotoxic T-cell lymphocyte antigen (CTLA) 4 antibodies, or anti-Programmed Death (Ligand)(PD)-(L)1 antibodies and improved clinical outcomes across various types of cancer [4,5,76,77,78,79,80,81,82,83].

Strikingly, little data exist on the study of B-Eos in NSCLC patients treated with ICI and outcomes (Table 1). The studies are all retrospective in nature. Authors noted a correlation between raised blood eosinophils and a favorable clinical or radiological outcome. The princeps study by Tanizaki and colleagues suggests a prognostic and/or predictive role of B-Eos in patients treated with nivolumab for advanced NSCLC after the failure of previous systemic treatment [5]. Pre-nivolumab absolute eosinophil count (AEC) >0.15 cells/mL, absolute lymphocyte count (ALC) >1.0 cells/mL, and absolute neutrophil count (ANC) >7.5 cells/mL were significantly associated with a better overall and progression-free survival (OS and PFS, respectively). This was confirmed in the tumors with PD-L1 expression ≥50% but was not significant for tumors with PD-L1 expression <50%. For patients with an AEC > 0.15 cells/mL, the risk of death was reduced by 76% and the risk of progression by 47%. Two other studies looking at leucocytes under ICI treatment comforted those results on a slightly higher number of patients and in a similar therapeutic context [82,83]. In our cohort of patients, none of the pre-treatment B-Eos values were predictive nor prognostic [82]. The relative eosinophil count (REC) was predictive of objective response according to the Response Criteria In Solid Tumor (RECIST) at the first evaluation [8–12 weeks after the first treatment] and at the second evaluation (+8–12 weeks) (*p* = 0.0019, OR = 0.54, and *p* = 0.0014, OR = 0.53, respectively). The duration of treatment, an indirect reflection of the clinical benefit, was significantly longer with a lower ANC (*p* = 0.0096) and a higher REC (*p* = 0.0021) at the first RECIST evaluation. Notably, no association was found between B-Eos and toxicity. Neutrophils, lymphocytes, and their ratio were prognostic in this treatment setting. Okauchi and colleagues concentrated on the study of B-Eos only [83]. They showed that pre-treatment AEC was lower in patients that would later progress under ICI (*p* = 0.002). Under treatment, AEC and REC were lower in progressive patients (*p* = 0.002 and <0.0001, respectively). The time to treatment failure was longer in patients with an AEC > 0.15 cells/mL and a REC > 3% before ICI initiation (*p* = 0.046 and 0.003, respectively) and with an AEC > 0.3 and >0.5 cells/mL (*p* < 0.001 for both) and a REC > 3 and >5% on treatment (*p* < 0.001 for both). The two latest studies further suggest, based on Receiver Operator Curves (ROC) analysis, that a REC > 5% is predictive of disease control, although with disputable sensitivity and specificity (81.9% and 32.8%, respectively [82]; 60.7% and 27.3%, respectively [83]). In the last study, Chu and colleagues analysed data from 300 NSCLC patients treated with ICI for advanced disease and looked at pre-treatment peripheral blood characteristics that may predict the occurrence of immune-related pneumonitis and predict general outcomes (survival and response rates) [81]. They demonstrated a link between pre-treatment AEC (cut-off value of 0.125 cells/mL) and [1] a higher objective response rate (ORR) [40.9% vs. 28.8%, *p* = 0.029] and [2] a longer PFS [8.9 vs. 5.9 months, *p* = 0.038].

As these data come from retrospective studies, the quality of the observations is clearly poorer. For instance, registration of medical conditions (allergy, asthma, COPD) and concomitant medications (corticoids) interfering with eosinophilia were only completely mentioned in one out of the four studies on NSCLC patients [82]. Additionally, the overview given in Table 1 allows considering the heterogeneity of the studies regarding the number of patients included and the evaluation criteria for B-Eos (studied as continuous vs. categorical variables; inconsistent evaluation time points; single vs. composite biomarker). However, there is a consistent correlation between raised B-Eos under treatment with ICI and better outcomes (OS, PFS, ORR).

Voorwerk and colleagues addressed the question of the specificity of ICI in inducing eosinophilia in their melanoma mouse model and demonstrated [1] that the rise in B-Eos after ICI was specific to this type of anti-neoplastic drug, as compared to chemotherapy, and that it also occurred when combining chemotherapy and ICI; [2] that the improved survival of mice treated with ICI relied upon eosinophils, as depletion of these cells by anti-Siglec8-antibodies resulted in survival that paralleled the survival of mice not treated with ICI. The results concerning raised B-Eos and clinical response were confirmed for metastatic bladder and lung cancer, as well as for early-stage mismatch repair proficient colon cancer [84]. To the best of our knowledge, there is also no clinical report pointing at a link between B-Eos or T-Eos and the efficacy of chemotherapy or tyrosine kinase inhibitors.

Blood eosinophilia has also been reported in cancer patients who display toxicity to ICI. So-called immune-related adverse events (irAE) are specific to these drugs and reflect excessive immune activation [85]. There are case reports as well as (mostly retrospective) studies showing an association between the occurrence of irAE and eosinophilia. In the context of NSCLC, the series of Chu et al revealed a correlation between baseline AEC and the occurrence of pneumonitis [27.7% if AEC ≥ 0.125 cells/mL vs. 9.8% if AEC < 0.125 cells/mL, *p* < 0.0001] [81].

Some authors advocate for the existence of a drug-driven, irAE-independent eosinophilic syndrome in the context of ICI [80,86]. Both groups demonstrated the existence of B-Eos (>0.5 cells/mL in Bernard–Tessier, >1.0 cells/mL in Scanvion) in the absence of irAE, although the retrospective nature of the study may not allow for a complete recording of toxicity events. However, the correlation between various drugs and eosinophilia is already well known and as such there is no reason that ICI could not lead to a similar phenomenon. In that case, the rise in eosinophils can be the consequence of increased production of these cells, e.g., IL-2 triggering IL-5 production, leading to increased eosinophilopoïesis, as observed in mouse models [87,88]. It can also be the result of a type IVb allergic reaction characterised by the occurrence of a Th2-mediated immune response, as seen in some patients taking various types of medication [89]. Given the wide clinical spectrum of medication-induced eosinophilia and the possible overlap of clinical signs with irAE (such as a rash), this drug-induced eosinophilia may, in fact, be underestimated.

### 5.2. Tissue Eosinophils (T-Eos)

To date, these data are scarce in NSCLC. In advanced disease, we found no report on tissue eosinophils (T-Eos) for this tumor type. In the early stages, two studies described eosinophils and their value in this setting. Ye and colleagues studied the expression of EPX, one of the four proteins contained in eosinophil granules, on 30 resection specimens of adenocarcinoma of the lung and on adjacent, normal lung tissue [90]. The expression level of EPX was rated by the degree (negative/weak/medium/strong staining) and the extent [0/1–25/26–50/51–75/76–100%] of the protein expression. A score was then defined for high vs. low EPX expression. Univariate analysis revealed a higher EPX expression in the cancer areas as compared with normal tissue (*p* < 0.05) and a correlation of high levels of EPX with higher pathological Tumor Node Metastases (pTNM) stage (*p* = 0.017) and with lymph node involvement (*p* = 0.027). T-Eos here was associated with a worse prognosis with a calculated hazard ratio (HR) for death of 3.1 (*p* = 0.018) in the EPX high group. Tataroglu and colleagues published a study on the presence of mast cells, macrophages, and eosinophils and their association with tumor vasculature and TNM stage NSCLC samples [91]. No significant association was noted between eosinophils and tumor stage or between tumor-associated vasculature and eosinophils. It should be noted, however, that eosinophils were evaluated by light microscopy after staining with hematoxylin-eosin. Weller and Spencer described the difficulties in detecting eosinophils in tissue thoroughly and suggested that electron microscopy or the use of antibodies directed at eosinophil granule proteins are useful tools to optimise the count of these cells in tissue [92]. In addition to technical issues, TATE could vary according to the degree of activation of the immune cascade, i.e., according to the interplay of cytokines, chemokines, and immune cells shaping the tumor microenvironment.

## 6. Perspectives

While clinical data suggest potential roles for eosinophils in NSCLC in the context of ICI treatment, preclinical models offer strong evidence that these myeloid cells do play an important role in the immune response against (lung) cancer. Furthermore, in vitro and animal models have revealed the complex interplay of different cells, whereof eosinophils, and components of the tumor microenvironment, leading to a priori opposed roles for eosinophils.

In order to further unravel the role of eosinophils in this context and, hence, to explore their possible predictive and/or prognostic value as biomarkers, it appears of fundamental importance to go over from descriptive findings, relying on the sole eosinophil count, to functional studies that will clarify what role(s) eosinophils fulfill in this setting. In asthma, those studies led to important advances in understanding their diversity and plasticity [71,73]. They showed that the role of resident eosinophils differs from those of inflammatory, allergy-induced eosinophils. Such functional studies, however, face technical challenges in humans. First, eosinophils are a numerically poorly represented myeloid cell population. Second, available techniques to access the functional repertoire of these cells, i.e., DNA, RNA, or proteins, all have their limitations and, until recently, rendered poor results, explaining the lack of functional characterisation data on human eosinophils, and in particular in lung cancer [93]. However, techniques are advancing fast and refinements have already made possible functional studies of mouse eosinophils [94].

Another issue that is yet to be solved is to strengthen the evidence from patient cohorts. Clearly, prospective data are needed to erase the biases inherent to the retrospective studies: incomplete data collection and the exclusion of patients based on a posteriori criteria. In particular, upfront registration of confounding factors such as concomitant medications (inhaled and systemic corticoids), known predictive factors of ICI efficacy (tumor PD-L1 and mutational status, smoking history, immune-related toxicity), or medical conditions (parasitic infections, atopy, asthma, COPD) is of paramount importance to ascertain (a) role(s) of eosinophils in lung cancer patients treated with ICI. Those roles, for now, can only be suggested based on the available data.

The variability of blood eosinophils is a well-known problem that may, at least in part, explain their lack of sensitivity in predicting clinical outcomes. It was formerly illustrated in the context of asthma and chronic obstructive pulmonary disease (COPD), where intra-patient, day-to-day variability but also circadian variability were demonstrated [95,96]. Given the lack of satisfying sensitivity in the two attempts to define a cut-off value for B-Eos to predict disease control in patients treated with ICI for NSCLC, the study of B-Eos should at least be challenged by studies on alternative materials. As lung cancer remains an air-borne disease, sputum, bronchoalveolar lavage, or exhaled air from lung cancer patients could provide useful information. Furthermore, although biopsies in lung cancer patients can be challenging and, in a substantial proportion of cases, will need invasive techniques, we feel that a baseline, i.e., pre-treatment, comparative assessment of eosinophils in tissue vs. other material would be valuable.

Once available, tissue should also be analysed with techniques offering the highest chance of locating (qualitative analysis) and counting (quantitative analysis) eosinophils as a first step. Such data are, at the present time, unavailable for advanced stages of NSCLC and are scarce for early stages. In any case, B-Eos and T-Eos potentially differ in terms of their ability to function, as they evolve in different conditions (such as the oxygen content). Therefore, a comparative study might be interesting. 

Arguably, one could wonder whether, given the difficulties summed up here, looking for the trigger of eosinophil activation (alarmins) would not be preferable to looking for the eosinophils themselves.

Another unexplored area in the clinical research described here is the study of the kinetics of blood eosinophilia. So far, only one study reported results for patients treated for multiple oncological indications with ICI [80]. While the rise in blood eosinophils is noted early in the treatment course, the study of their evolution over time could provide valuable observations to guide further clinical and/or translational research.

## 7. Conclusions

Preclinical models have established a role, although not unique, for tissue eosinophils in cancer. Despite their questionable quality, clinical data suggest that raised blood eosinophils may reflect a favorable outcome in patients treated with immune checkpoint inhibitors for advanced NSCLC. Functional studies and more stringent clinical research are needed to further elucidate the role of eosinophils in lung cancer and their potential value as a biomarker.

## Figures and Tables

**Figure 1 ijms-23-05066-f001:**
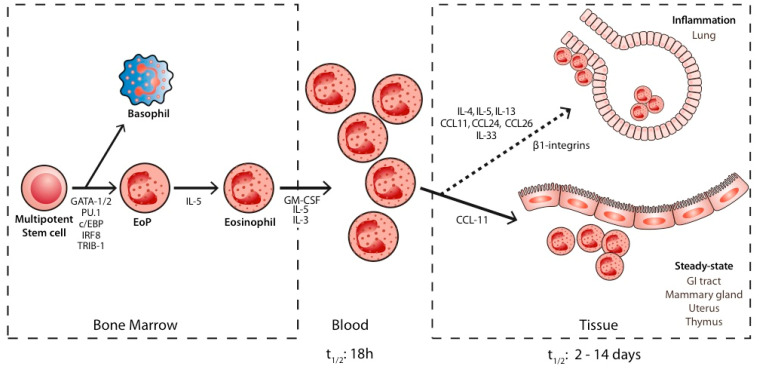
Biology of eosinophils. Eosinophils derive from multipotent stem cells. They proliferate, migrate, and are activated by cytokines, mainly Interleukin-5 (IL-5). They spend a short time in blood and subsequently migrate to tissues via the interplay of several chemokines. GM-CSF: Granulocyte-Macrophage–Colony Stimulating Factor. EoP: eosinophil progrenitor. IL-5: Interleukin-5. IL-3: Interleukin-3; CCL11: CC-chemokine ligand 11(=eotaxin1); CCL24: eotaxin-2; CCL-16: eotaxin-3. T^1/2^: half-life. GI tract: gastrointestinal tract.

**Figure 2 ijms-23-05066-f002:**
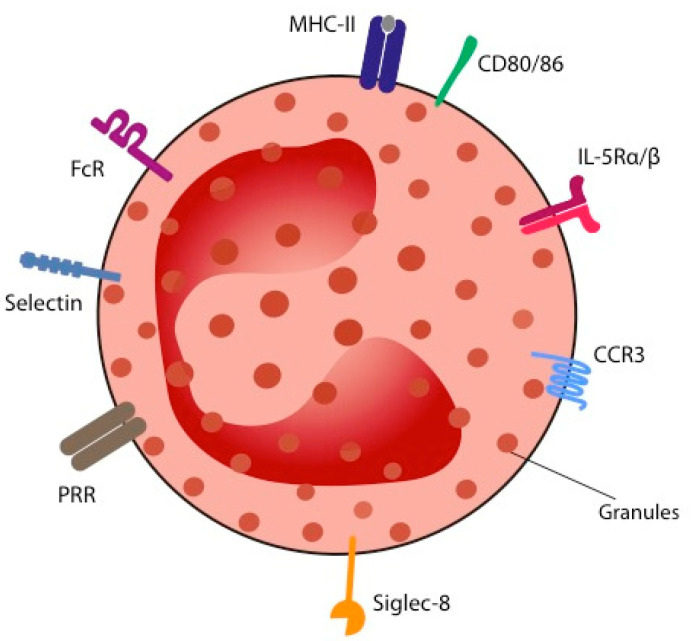
Structure of the human eosinophil. Eosinophils can be characterised by their surface markers and by their intracellular content. Cell-surface markers are: adhesion molecules (selectins) allowing for adhesion and endothelial transmigration; chemokine receptors (CCR) and chemotactic factors allowing for the attraction and local activation of eosinophils; cytokine and growth factor receptors (e.g., Interleukin-5 Receptor alpha subunit (IL-5Rα)); complement receptors; immunoglobulin receptors (e.g., FcR); inhibitory receptors (e.g., Sialic acid-binding immunoglobulin-like lectin-8 (Siglec-8)) and pattern recognition receptors (PRR), e.g., Toll-like receptors whose activation is triggered by alarmins (Pathogen-associated molecular patterns (PAMPs) in case of infection and Danger-associated molecular patterns (DAMPs) in case of tumor).

**Figure 3 ijms-23-05066-f003:**
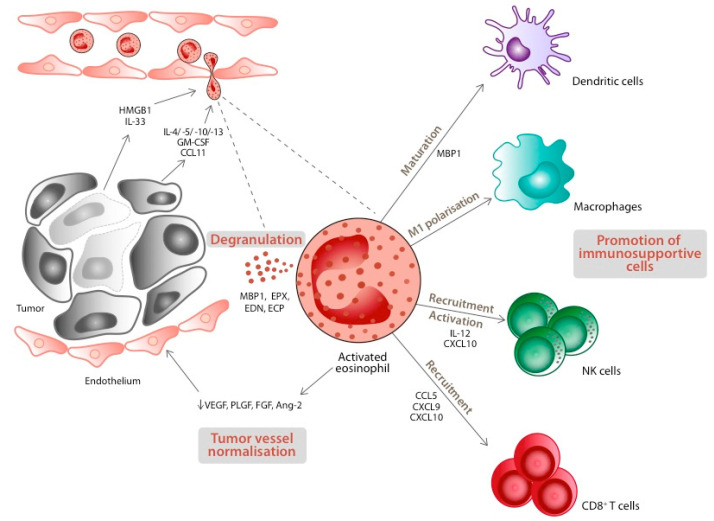
Eosinophil recruitment at tumor sites and anti-tumor effects of eosinophils. In response to their recruitment and activation via different cytokines and chemokines such as tumor-secreted Interleukin-5 (IL-5), or IL-33 and High Mobility Group Box-1 protein (HMGB-1), alarmins secreted by dying tumor cells, eosinophils display both direct and indirect anti-tumorigenic effects. Degranulation of eosinophils has cytotoxic and ribonucleasic effects. Moroever, activated eosinophils are capable of recruiting immune cells to engage against tumors: Natural Killer (NK) cells, cytotoxic CD8^+^ T cells, and dendritic cells (DC). Additionally, they can polarise macrophages to an M1, anti-tumorigenic phenotype. Finally, eosinophils appear to affect tumor vasculature by increasing vascular leakiness, leading to tumor necrosis. IL: Interleukin; HMGB-1: High Mobility Group Box-1 protein; PRR: Pattern Recognition Receptor; CCL11: CC-chemokine ligand 11 = eotaxin1; CXCL9: CXC-chemokine ligand 9; MBP-1: major basic protein-1; EPX: eosinophil peroxidase; EDN: eosinophil-derived neurotoxin; ECP: eosinophil cationic protein; ↓: reduced expression; VEGF: vascular endothelial growth factor; PLGF: platelet growth factor; FGF: fibroblast growth factor; Ang-2: angiopoietin-2.

**Table 1 ijms-23-05066-t001:** Studies on the association between outcomes of NSCLC patients treated with ICI and blood eosinophils. This table illustrates the heterogeneity of study objectives and of evaluation criteria for eosinophilia: continuous/categorical variable; timing of evaluation; biomarker used alone (simple) or in combination with others (composite).

Study	N	Stage of Disease	ICI	Eosinophils	Outcome	Effects	*p* Value
Tanizaki 2017 [5]	134	IIIB-IV	nivolumab	AEC t0; categorical; simple & composite biomarker	OS PFS	HR = 0.24 [95% CI 0.09−0.62] HR = 0.53 [95% CI 0.31−0.91] if AECt0 ≥ 0.15 cells/mL	0.003 0.02
Chu X 2020 [81]	300	IIIB-IV	PD-1i +/− CT +/− AAG	AEC t0; categorical; simple	ORR PFS	40.9 % vs 28.8 % med. = 8.93 vs 5.87 mo HR = 0.744 [95% CI 0.56−0.99] if AECt0 ≥ 0.15 cells/mL	0.029 0.038
Sibille 2021 [82]	191	IIIA-IV	pembrolizumab nivolumab atezolizumab durvalumab	AEC & REC t1; continuous	ORR	OR = 0.53 [95% CI 0.32−0.88] if RECt1 > 5.3%	0.014
Okauchi 2021 [83]	190	IIIA-IV	nivolumab pembrolizumab atezolizumab +/− CT	AEC & REC t0 & q2–3 wk; RECmax. *; categorical	TTF	OR = 0.39 [95% CI 0.26−0.60] if RECmax. > 5%	<0.001

ICI: immune checkpoint inhibitor; NSCLC: non-small cell lung cancer; PD-1i: Programmed death-1 inhibitor; AAG: anti-angiogenics; CT: chemotherapy (platinum-based doublet); AEC: absolute eosinophil count; REC: relative eosinophil count; categorical: studied as a categorical variable; continuous: studied as a continuous variable; t0: value before ICI treatment; t1: timing of the first RECIST evaluation under ICI treatment (at 8–12 weeks after initiation); q2–3 wk: every 2–3 weeks; * REC max.: maximal REC value noted under ICI; OS: overall survival; PFS: progression-free survival; ORR: objective response rate; TTF: time to treatment failure; CI: confidence interval; HR: hazard ratio; OR: odds ratio.

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
