# Peer review of "Eosinophils and Lung Cancer: From Bench to Bedside"

_ijms, 2022, doi:10.3390/ijms23095066_

Round 1
Reviewer 1 Report
Compared to the original version, the quality and clarity of the manuscript has been considerably improved and now the papers deserves the publication in IJMS. I would only suggest to shortly summarise the scope and take-home message of the paper at the end of the Abstract (see the last section of Introduction). Next, the Authors could consider the discussion of the data from blood samples before referring to tissue samples.
Reviewer 2 Report
None
Author Response
Please see the attachment.

This manuscript is a resubmission of an earlier submission. The following is a list of the peer review reports and author responses from that submission.
Round 1
Reviewer 1 Report
The paper focuses on the role of eosinophils in the systemic immune response to cancer, with an emphasis to their pro- and anti-tumorigenic effects in lung cancer. Then it mentions the challenges for a better understanding of the role of eosinophils in lung cancer. The paper is relatively well written and has a clear structure. However, there are several critical points that raised my scepticism about this manuscript, in particular:
- its rational for the paper and its message is somewhat opaque: the Authors start with a comprehensive description of systemic eosinophil function, then they focus on relatively short and inconclusive list of studies on the role of eosinophils in tumors in general and particularly in lung cancer, finishing with rather obvious and superficial conclusion on the need for further research on this topic. Collectively, there is no “take-home message”, which prompts the impression that the publication of the paper may be premature;
- perhaps the results of in vivo/in vitro studies on the pro-/anti-tumorigenic function of eosinophil effectors, such as macrophages, would help to enhance the impact of the paper;
- consequently, an outline of controversial role of inflammation in cancer could provide a context and would also help the reader to understand the principles of multidirectional function of eosinophils;
- “the importance of the tumor microenvironment in orientating the role of eosinophils” is mentioned in the Abstract but hardly covered in the text. The Authors could concentrate on this point to improve the message of the story;
Other points:
- Abstract: Review focus/rational should be provided in the last sentences
- haematopoietic stem cells are multipotent rather than pluripotent;
- “patients displaying the toxicity to drugs”: is it correct?
- selectins are lacking in the Fig. 2;
- 4.3. lacks continuity and is difficult to follow.
Reviewer 2 Report
The authors reviewed here the role of eosinophils in lung cancer, but the discussion of eosinophilia in lung cancer is limited to 1 page, since there is a limited body of evidence in literature on eosinophils and lung cancer.
In my experience, the presence of eosinophils in lung cancer is a finding related to poor prognosis. However, the cases collected are too few to consistently demonstrate a real value (negative or positive) of eosinophilia in lung cancer.
Another missed point is the distinction between tissutal and serologic eosinophilia (or even combined). In the paper, the authors did not focus on this finding that could be very important to implement the number of papers entering in discussion.
Tissutal or serum eosinophilia before and after treatment or based on different treatments (e.g., chemotherapy, immunotherapy, TKIs) or in naive lung cancer is missed.
Reviewer 3 Report
Major points:
- The review article is poorly written. There are many typos and grammar errors in the present manuscript. The present manuscript needs professional English editing.
- Table 1. Please enroll NSCLC studies only and make more detailed information, the regimens, basic characteristics and outcome, irAE, and so on. CTLA4 and PDL-1 inhibitors include several ICIs, authors need to present a comprehensive review of the relationship between different ICIs and eosinophils in the treatment of NSCLC clinically.
- Figure 3 should be re-plotted. The authors should point out the recruitment or differentiation or activation of the immune cells from eosinophils.
- References should be up-to-date. The papers are less than 40% published within the recent 5 years.
Minor points:
- The format of references should fit the style of this journal.
- The Ref #14, #35, #38, #50 and #81 are wrong. The authors should correct them.